# Adversarial Multi-task Learning for Text Classification

## Abstract

Neural network models have shown their promising opportunities for multi-task learning, which focus on learning the shared layers to extract the common and task-invariant features. However, in most existing approaches, the extracted shared features are prone to be contaminated by task-specific features or the noise brought by other tasks. In this paper, we propose an adversarial multi-task learning framework, alleviating the shared and private latent feature spaces from interfering with each other. We conduct extensive experiments on 16 different text classification tasks, which demonstrates the benefits of our approach. Besides, we show that the shared knowledge learned by our proposed model can be regarded as off-the-shelf knowledge and easily transferred to new tasks.

## 1 Introduction

Multi-task learning is an effective approach to improve the performance of a single task with the help of other related tasks. Recently, neural-based models for multi-task learning have become very popular, ranging from computer vision (Misra et al., 2016; Zhang et al., 2014) to natural language processing (Collobert and Weston, 2008; Luong et al., 2015), since they provide a convenient way of combining information from multiple tasks.

However, most existing work on multi-task learning attempts to divide the features of different tasks into private and shared spaces, merely based on whether parameters of some components should be shared. As shown in Figure 1-(a), the general shared-private model introduces two feature spaces for any task: one is used to

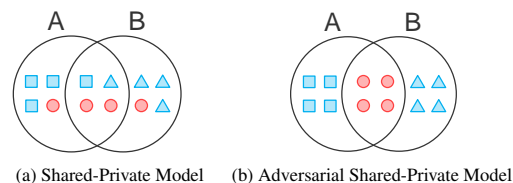

Figure 1: Two sharing schemes for task A and task B. The overlap between two black circles denotes shared space. The blue triangles and boxes represent the task-specific features while the red circles denote the features which can be shared.

store task-dependent features, the other is used to capture shared features. The major limitation of this framework is that the shared feature space could contain some unnecessary task-specific features, while some sharable features could also be mixed in private space, suffering from feature redundancy.

Taking the following two sentences as examples, which are extracted from two different sentiment classification tasks: Movie reviews and Baby products reviews.

*The **infantile** cart is simple and easy to use.*
*This kind of humour is **infantile** and boring.*

The word "`infantile`" indicates negative sentiment in Movie task while it is neutral in Baby task. However, the general shared-private model could place the task-specific word "`infantile`" in a shared space, leaving potential hazards for other tasks. Additionally, the capacity of shared space could also be wasted by some unnecessary features.

To address this problem, in this paper we propose an adversarial multi-task framework, in which the shared and private feature spaces are inherently disjoint by introducing orthogonality constraints. Specifically, we design a generic shared-private learning framework to model the text se-

quence. To prevent the shared and private latent feature spaces from interfering with each other, we introduce two strategies: adversarial training and orthogonality constraints. The adversarial training is used to ensure that the shared feature space simply contains common and task-invariant information, while the orthogonality constraint is used to eliminate redundant features from the private and shared spaces.

The contributions of this paper can be summarized as follows.

1. Proposed model divides the task-specific and shared space in a more precise way, rather than roughly sharing parameters.
2. We extend the original binary adversarial training to multi-class, which not only enables multiple tasks to be jointly trained, but allows us to utilize unlabeled data.
3. We can condense the shared knowledge among multiple tasks into an off-the-shelf neural layer, which can be easily transferred to new tasks.

## 2 Recurrent Models for Text Classification

There are many neural sentence models, which can be used for text modelling, involving recurrent neural networks (Sutskever et al., 2014; Chung et al., 2014), convolutional neural networks (Collobert et al., 2011; Kalchbrenner et al., 2014), and recursive neural networks (Socher et al., 2013). Here we adopt recurrent neural network with long short-term memory (LSTM) due to their superior performance in various NLP tasks.

**Long Short-term Memory** Long short-term memory network (LSTM) (Hochreiter and Schmidhuber, 1997) is a type of recurrent neural network (RNN) (Elman, 1990), and specifically addresses the issue of learning long-term dependencies. While there are numerous LSTM variants, here we use the LSTM architecture used by (Jozefowicz et al., 2015), which is similar to the architecture of (Graves, 2013) but without peep-hole connections.

We define the LSTM *units* at each time step $t$ to be a collection of vectors in $\mathbb{R}^d$: an *input gate* $\mathbf{i}_t$, a *forget gate* $\mathbf{f}_t$, an *output gate* $\mathbf{o}_t$, a *memory cell* $\mathbf{c}_t$ and a hidden state $\mathbf{h}_t$. $d$ is the number of the LSTM units. The elements of the gating vectors $\mathbf{i}_t$, $\mathbf{f}_t$ and $\mathbf{o}_t$ are in $[0, 1]$.

The LSTM is precisely specified as follows.

$$\begin{bmatrix} \tilde{\mathbf{c}}_t \\ \mathbf{o}_t \\ \mathbf{i}_t \\ \mathbf{f}_t \end{bmatrix} = \begin{bmatrix} \tanh \\ \sigma \\ \sigma \\ \sigma \end{bmatrix} \left( \mathbf{W}_p \begin{bmatrix} \mathbf{x}_t \\ \mathbf{h}_{t-1} \end{bmatrix} + \mathbf{b}_p \right), \quad (1)$$

$$\mathbf{c}_t = \tilde{\mathbf{c}}_t \odot \mathbf{i}_t + \mathbf{c}_{t-1} \odot \mathbf{f}_t, \quad (2)$$

$$\mathbf{h}_t = \mathbf{o}_t \odot \tanh(\mathbf{c}_t), \quad (3)$$

where $\mathbf{x}_t \in \mathbb{R}^e$ is the input at the current time step; $\mathbf{W}_p \in \mathbb{R}^{4d \times (d+e)}$ and $\mathbf{b}_p \in \mathbb{R}^{4d}$ are parameters of affine transformation; $\sigma$ denotes the logistic sigmoid function and $\odot$ denotes elementwise multiplication.

The update of each LSTM unit can be written precisely as follows:

$$\mathbf{h}_t = \mathbf{LSTM}(\mathbf{h}_{t-1}, \mathbf{x}_t, \theta_p). \quad (4)$$

Here, the function $\mathbf{LSTM}(\cdot, \cdot, \cdot, \cdot)$ is a shorthand for Eq. (1-3), and $\theta_p$ represents all the parameters of LSTM.

**Text Classification with LSTM** Given a text sequence $x = \{x_1, x_2, \cdots, x_T\}$, we first use a lookup layer to get the vector representation (embeddings) $\mathbf{x}_i$ of the each word $x_i$. The output at the last moment $\mathbf{h}_T$ can be regarded as the representation of the whole sequence, which has a fully connected layer followed by a softmax non-linear layer that predicts the probability distribution over classes.

$$\hat{\mathbf{y}} = \mathrm{softmax}(\mathbf{W}\mathbf{h}_T + \mathbf{b}) \quad (5)$$

where $\hat{\mathbf{y}}$ is prediction probabilities, $\mathbf{W}$ is the weight which needs to be learned, $\mathbf{b}$ is a bias term.

Given a corpus with $N$ training samples $(x_i, y_i)$, the parameters of the network are trained to minimise the cross-entropy of the predicted and true distributions.

$$L(\hat{y}, y) = -\sum_{i=1}^{N} \sum_{j=1}^{C} y_i^j \log(\hat{y}_i^j), \quad (6)$$

where $y_i^j$ is the ground-truth label; $\hat{y}_i^j$ is prediction probabilities, and $C$ is the class number.

## 3 Multi-task Learning for Text Classification

The goal of multi-task learning is to utilizes the correlation among these related tasks to improve

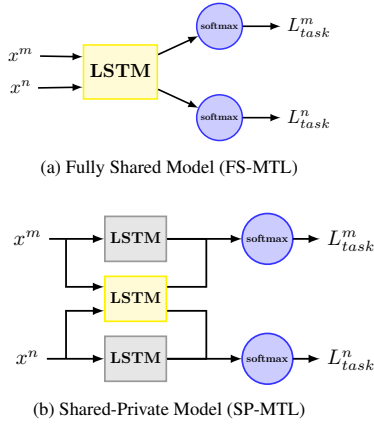

(a) Fully Shared Model (FS-MTL)

(b) Shared-Private Model (SP-MTL)

Figure 2: Two architectures for learning multiple tasks. Yellow and gray boxes represent shared and private LSTM layers respectively.

classification by learning tasks in parallel. To facilitate this, we give some explanation for notations used in this paper. Formally, we refer to $D_k$ as a dataset with $N_k$ samples for task $k$. Specifically,

$$D_k = \{(x_i^k, y_i^k)\}_{i=1}^{N_k} \qquad (7)$$

where $x_i^k$ and $y_i^k$ denote a sentence and corresponding label for task $k$.

### 3.1 Two Sharing Schemes for Sentence Modeling

The key factor of multi-task learning is the sharing scheme in latent feature space. In neural network based model, the latent features can be regarded as the states of hidden neurons. Specific to text classification, the latent features are the hidden states of LSTM at the end of a sentence. Therefore, the sharing schemes are different in how to group the shared features. Here, we first introduce two sharing schemes with multi-task learning: fully-shared scheme and shared-private scheme.

**Fully-Shared Model (FS-MTL)**  In fully-shared model, we use a single shared LSTM layer to extract features for all the tasks. For example, given two tasks $m$ and $n$, it takes the view that the features of task $m$ can be totally shared by task $n$ and vice versa. This model ignores the fact that some features are task-dependent. Figure 2a illustrates the fully-shared model.

**Shared-Private Model (SP-MTL)**  As shown in Figure 2b, the shared-private model introduces two feature spaces for each task: one is used to store task-dependent features, the other is used

to capture task-invariant features. Accordingly, we can see each task is assigned a private LSTM layer and shared LSTM layer. Formally, for any sentence in task $k$, we can compute its shared representation $\mathbf{s}_t^k$ and task-specific representation $\mathbf{h}_t^k$ as follows:

$$\mathbf{s}_t^k = \mathbf{LSTM}(x_t, \mathbf{s}_{t-1}^k, \theta_s), \qquad (8)$$
$$\mathbf{h}_t^k = \mathbf{LSTM}(x_t, \mathbf{h}_{t-1}^m, \theta_k) \qquad (9)$$

where $\mathbf{LSTM}(., \theta)$ is defined as Eq. (4).

The final features are concatenation of the features from private space and shared space.

### 3.2 Task-Specific Output Layer

For a sentence in task $k$, its feature $\mathbf{h}^{(k)}$, emitted by the deep muti-task architectures, is ultimately fed into the corresponding task-specific softmax layer for classification or other tasks.

The parameters of the network are trained to minimise the cross-entropy of the predicted and true distributions on all the tasks. The loss $L_{task}$ can be computed as:

$$L_{Task} = \sum_{k=1}^{K} \alpha_k L(\hat{y}^{(k)}, y^{(k)}) \qquad (10)$$

where $\alpha_k$ is the weights for each task $k$ respectively. $L(\hat{y}, y)$ is defined as Eq. 6.

## 4 Incorporating Adversarial Training

Although the shared-private model separates the feature space into the shared and private spaces, there is no guarantee that sharable features can not exist in private feature space, or vice versa. Thus, some useful sharable features could be ignored in shared-private model, and the shared feature space is also vulnerable to contamination by some task-specific information.

Therefore, a simple principle can be applied into multi-task learning that a good shared feature space should contain more common information and no task-specific information. To address this problem, we introduce adversarial training into multi-task framework as shown in Figure 3 (ASP-MTL).

### 4.1 Adversarial Network

Adversarial networks have recently surfaced and are first used for generative model (Goodfellow

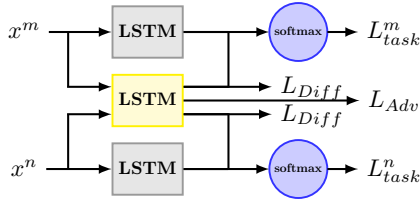

Figure 3: Adversarial shared-private model. Yellow and gray boxes represent shared and private LSTM layers respectively.

et al., 2014). The goal is to learn a generative distribution $p_G(x)$ that matches the real data distribution $P_{data}(x)$ Specifically, GAN learns a generative network G and discriminative model D, in which G generates samples from the generator distribution $p_G(x)$. and D learns to determine whether a sample is from $p_G(x)$ or $P_{data}(x)$. This min-max game can be optimized by the following risk:

$$\phi = \min_G \max_D \Big( E_{x \sim P_{data}}[\log D(x)]$$
$$+ E_{z \sim p(z)}[\log(1 - D(G(z)))] \Big) \quad (11)$$

While originally proposed for generating random samples, adversarial network can be used as a general tool to measure equivalence between distributions (Taigman et al., 2016). Formally, (Ajakan et al., 2014) linked the adversarial loss to the $H$-divergence between two distributions and successfully achieve unsupervised domain adaptation with adversarial network. Motivated by theory on domain adaptation (Ben-David et al., 2010, 2007; Bousmalis et al., 2016) that a transferable feature is one for which an algorithm cannot learn to identify the domain of origin of the input observation.

## 4.2 Task Adversarial Loss for MTL

Inspired by adversarial networks (Goodfellow et al., 2014), we proposed an adversarial shared-private model for multi-task learning, in which a shared recurrent neural layer is working adversarially towards a learnable multi-layer perceptron, preventing it from making an accurate prediction about the types of tasks. This adversarial training encourages shared space to be more pure and ensure the shared representation not be contaminated by task-specific features.

**Task Discriminator**  Discriminator is used to map the shared representation of sentences into a

probability distribution, estimating what kinds of tasks the encoded sentence comes from.

$$D(\mathbf{s}_T^k, \theta_D) = \text{softmax}(\mathbf{b} + \mathbf{U}\mathbf{s}_T^k) \quad (12)$$

where $\mathbf{U} \in \mathbb{R}^{d \times d}$ is a learnable parameter and $\mathbf{b} \in \mathbb{R}^d$ is a bias.

**Adversarial Loss**  Different with most existing multi-task learning algorithm, we add an extra task adversarial loss $L_{Adv}$ to prevent task-specific feature from creeping in to shared space. The task adversarial loss is used to train a model to produce shared features such that a classifier cannot reliably predict the task based on these features. The original loss of adversarial network is limited since it can only be used in binary situation. To overcome this, we extend it to multi-class form, which allow our model can be trained together with multiple tasks:

$$L_{Adv} = \min_{\theta_s} \left( \lambda \max_{\theta_D} (\sum_{k=1}^{K} \sum_{i=1}^{N_k} d_i^k \log[D(E(\mathbf{x}^k))]) \right) \quad (13)$$

where $d_i^k$ denotes the ground-truth label indicating the type of the current task. Here, there is a min-max optimization and the basic idea is that, given a sentence, the shared LSTM generates a representation to mislead the task discriminator. At the same time, the discriminator tries its best to make a correct classification on the type of task. After the training phase, the shared feature extractor and task discriminator reach a point at which both cannot improve and the discriminator is unable to differentiate among all the tasks.

**Semi-supervised Learning Multi-task Learning**  We notice that the $L_{Adv}$ requires only the input sentence $x$ and does not require the corresponding label $y$, which makes it possible to combine our model with semi-supervised learning. Finally, in this semi-supervised multi-task learning framework, our model can not only utilize the data from related tasks, but can employ abundant unlabeled corpora.

## 4.3 Orthogonality Constraints

We notice that there is a potential drawback of the above model. That is, the task-invariant features can appear both in shared space and private space.

Motivated by recently work(Jia et al., 2010; Salzmann et al., 2010; Bousmalis et al., 2016)

| Dataset | Train | Dev. | Test | Unlab. | Avg. L | Vocab. |
|---------|-------|------|------|--------|--------|--------|
| Books   | 1400  | 200  | 400  | 2000   | 159    | 62K    |
| Elec.   | 1398  | 200  | 400  | 2000   | 101    | 30K    |
| DVD     | 1400  | 200  | 400  | 2000   | 173    | 69K    |
| Kitchen | 1400  | 200  | 400  | 2000   | 89     | 28K    |
| Apparel | 1400  | 200  | 400  | 2000   | 57     | 21K    |
| Camera  | 1397  | 200  | 400  | 2000   | 130    | 26K    |
| Health  | 1400  | 200  | 400  | 2000   | 81     | 26K    |
| Music   | 1400  | 200  | 400  | 2000   | 136    | 60K    |
| Toys    | 1400  | 200  | 400  | 2000   | 90     | 28K    |
| Video   | 1400  | 200  | 400  | 2000   | 156    | 57K    |
| Baby    | 1300  | 200  | 400  | 2000   | 104    | 26K    |
| Mag.    | 1370  | 200  | 400  | 2000   | 117    | 30K    |
| Soft.   | 1315  | 200  | 400  | 2000   | 129    | 26K    |
| Sports  | 1400  | 200  | 400  | 2000   | 94     | 30K    |
| IMDB    | 1400  | 200  | 400  | 2000   | 269    | 44K    |
| MR      | 1400  | 200  | 400  | 2000   | 21     | 12K    |

Table 1: Statistics of the 16 datasets. The columns 2-5 denote the number of samples in training, development, test and unlabeled sets. The last two columns represent the average length and vocabulary size of corresponding dataset.

on shared-private latent space analysis, we introduce orthogonality constraints, which penalize redundant latent representations and encourages the shared and private extractors to encode different aspects of the inputs.

After exploring many optional methods, we find below loss is optimal, which is used by Bousmalis et al. (2016) and achieve a better performance:

$$L_{\text{diff}} = \sum_{k=1}^{K} \left\| \mathbf{S}^{k\top} \mathbf{H}^k \right\|_F^2 , \qquad (14)$$

where $\| \cdot \|_F^2$ is the squared Frobenius norm. $\mathbf{S}^k$ and $\mathbf{H}^k$ are two matrics, whose rows are the output of shared extractor $E_s(,;\theta_s)$ and task-specific extrator $E_k(,;\theta_k)$ of a input sentence.

## 4.4 Put It All Together

The final loss function of our model can be written as:

$$L = L_{Task} + \lambda L_{Adv} + \gamma L_{Diff} \qquad (15)$$

where $\lambda$ and $\gamma$ are hyper-parameter.

The networks are trained with backpropagation and this minimax optimization becomes possible via the use of a gradient reversal layer (Ganin and Lempitsky, 2015).

## 5 Experiment

### 5.1 Dataset

To make an extensive evaluation, we collect 16 different datasets from several popular review corpora.

The first 14 datasets are product reviews, which contain Amazon product reviews from different domains, such as Books, DVDs, Electronics, ect. The goal is to classify a product review as either positive or negative. These datasets are collected based on the raw data [1] provided by (Blitzer et al., 2007). Specifically, we extract the sentences and corresponding labels from the unprocessed original data [2]. The only preprocessing operation of these sentences is tokenized using the Stanford tokenizer [3].

The remaining two datasets are about movie reviews. The IMDB dataset[4] consists of movie reviews with binary classes (Maas et al., 2011). One key aspect of this dataset is that each movie review has several sentences. The MR dataset also consists of movie reviews from rotten tomato website with two classes [5](Pang and Lee, 2005).

All the datasets in each task are partitioned randomly into training set, development set and testing set with the proportion of 70%, 20% and 10% respectively. The detailed statistics about all the datasets are listed in Table 1.

### 5.2 Competitor Methods for Multi-task Learning

The multi-task frameworks proposed by previous works are various while not all can be applied to the tasks we focused. Nevertheless, we chose two most related neural models for multi-task learning and implement them as competitor methods.

- MT-CNN: This model is proposed by Collobert and Weston (2008) with convolutional layer, in which lookup-tables are shared partially while other layers are task-specific.

---

[1] https://www.cs.jhu.edu/~mdredze/datasets/sentiment/
[2] Blitzer et al. (2007) also provides two extra processed datasets with the format of Bag-of-Words, which are not proper for neural-based models.
[3] http://nlp.stanford.edu/software/tokenizer.shtml
[4] https://www.cs.jhu.edu/~mdredze/datasets/sentiment/unprocessed.tar.gz
[5] https://www.cs.cornell.edu/people/pabo/movie-review-data/.

| Task | Single Task | | | | Multiple Tasks | | | | |
|------|------|--------|-------|------|--------|--------|--------|--------|---------|
| | LSTM | BiLSTM | sLSTM | Avg. | MT-DNN | MT-CNN | FS-MTL | SP-MTL | ASP-MTL |
| Books | 20.5 | 19.0 | 18.0 | 19.2 | $17.7_{(-1.5)}$ | $15.6_{(-3.6)}$ | $17.5_{(-1.7)}$ | $18.7_{(-0.5)}$ | $13.0_{(-6.2)}$ |
| Electronics | 19.5 | 21.5 | 23.3 | 21.4 | $18.2_{(-3.2)}$ | $16.9_{(-4.5)}$ | $14.3_{(-7.1)}$ | $12.3_{(-9.1)}$ | $11.0_{(-10.4)}$ |
| DVD | 18.3 | 19.5 | 22.0 | 19.9 | $15.8_{(-4.1)}$ | $16.1_{(-3.8)}$ | $16.5_{(-3.4)}$ | $16.1_{(-3.8)}$ | $12.6_{(-7.3)}$ |
| Kitchen | 22.0 | 18.8 | 19.5 | 20.1 | $19.2_{(-0.9)}$ | $16.8_{(-3.3)}$ | $14.0_{(-6.1)}$ | $14.8_{(-5.3)}$ | $12.8_{(-7.3)}$ |
| Apparel | 16.8 | 14.0 | 16.3 | 15.7 | $14.9_{(-0.8)}$ | $16.1_{(+0.4)}$ | $15.5_{(-0.2)}$ | $13.4_{(-2.3)}$ | $11.3_{(-4.4)}$ |
| Camera | 14.8 | 14.0 | 15.0 | 14.6 | $13.7_{(-0.9)}$ | $14.0_{(-0.6)}$ | $13.5_{(-1.1)}$ | $12.1_{(-2.5)}$ | $8.7_{(-5.9)}$ |
| Health | 15.5 | 21.3 | 16.5 | 17.8 | $14.3_{(-3.5)}$ | $12.9_{(-4.9)}$ | $12.0_{(-5.8)}$ | $12.8_{(-5.0)}$ | $10.9_{(-6.9)}$ |
| Music | 23.3 | 22.8 | 23.0 | 23.0 | $15.3_{(-7.7)}$ | $16.3_{(-6.7)}$ | $18.8_{(-4.2)}$ | $17.0_{(-6.0)}$ | $17.4_{(-5.6)}$ |
| Toys | 16.8 | 15.3 | 16.8 | 16.3 | $12.1_{(-4.2)}$ | $10.9_{(-5.4)}$ | $15.5_{(-0.8)}$ | $14.9_{(-1.4)}$ | $11.2_{(-5.1)}$ |
| Video | 18.5 | 16.3 | 16.3 | 17.0 | $15.0_{(-2.0)}$ | $18.7_{(+1.7)}$ | $16.3_{(-0.7)}$ | $16.8_{(-0.2)}$ | $14.5_{(-2.5)}$ |
| Baby | 15.3 | 16.5 | 15.8 | 15.9 | $12.1_{(-3.8)}$ | $12.4_{(-3.5)}$ | $12.0_{(-3.9)}$ | $13.2_{(-2.7)}$ | $10.2_{(-5.7)}$ |
| Magazines | 10.8 | 8.5 | 12.3 | 10.5 | $10.6_{(+0.1)}$ | $12.3_{(+1.8)}$ | $7.5_{(-3.0)}$ | $8.1_{(-2.4)}$ | $7.6_{(-2.9)}$ |
| Software | 15.3 | 14.3 | 14.5 | 14.7 | $14.4_{(-0.3)}$ | $13.4_{(-1.3)}$ | $13.8_{(-0.9)}$ | $13.1_{(-1.6)}$ | $12.7_{(-2.0)}$ |
| Sports | 18.3 | 16.0 | 17.5 | 17.3 | $16.8_{(-0.5)}$ | $16.1_{(-1.2)}$ | $14.5_{(-2.8)}$ | $12.7_{(-4.6)}$ | $13.3_{(-4.0)}$ |
| IMDB | 18.3 | 15.0 | 18.5 | 17.3 | $16.7_{(-0.6)}$ | $13.7_{(-3.6)}$ | $17.5_{(+0.2)}$ | $15.2_{(-2.1)}$ | $14.2_{(-3.1)}$ |
| MR | 27.3 | 25.3 | 28.0 | 26.9 | $24.5_{(-2.4)}$ | $25.5_{(-1.4)}$ | $25.3_{(-1.6)}$ | $24.1_{(-2.8)}$ | $22.7_{(-4.2)}$ |
| AVG | 18.2 | 17.4 | 18.3 | 18.0 | $15.7_{(-2.3)}$ | $15.5_{(-2.5)}$ | $15.3_{(-2.7)}$ | $14.7_{(-3.3)}$ | $12.8_{(-5.2)}$ |

Table 2: Error rates of our models on 16 datasets against typical baselines. The numbers in brackets represent the improvements relative to the average performance (Avg.) of three single task baselines.

- MT-DNN: The model is proposed by Liu et al. (2015) with bag-of-words input and multi-layer perceptrons, in which a hidden layer is shared.

## 5.3 Hyperparameters

The word embeddings for all of the models are initialized with the 200d GloVe vectors (840B token version, (Pennington et al., 2014)). The other parameters are initialized by randomly sampling from uniform distribution in $[-0.1, 0.1]$. The mini-batch size is set to 16.

For each task, we take the hyperparameters which achieve the best performance on the development set via an small grid search over combinations of the initial learning rate $[0.1, 0.01]$, $\lambda \in [0.01, 0.1]$, and $\gamma \in [0.01, 0.1]$. Finally, we chose the learning rate as 0.01, $\lambda$ as 0.05 and $\gamma$ as 0.01.

## 5.4 Performance Evaluation

Table 2 shows the error rates on 16 text classification tasks. The column of "Single Task" shows the results of vanilla LSTM, bidirectional LSTM (BiLSTM), stacked LSTM (sLSTM) and the average error rates of previous three models. The column of "Multiple Tasks" shows the results achieved by corresponding multi-task models. From this table, we can see that the performance of most tasks can be improved with a large margin with the help of multi-task learning, in which our model achieves the lowest error rates.

More concretely, compared with SP-MTL, ASP-MTL achieves $5.2\%$ average improvement surpassing SP-MTL with $1.9\%$, which indicates the importance of adversarial learning. It is noteworthy that for FS-MTL, the performances of some tasks are degraded, since this model puts all private and shared information into a unified space.

## 5.5 Shared Knowledge Transfer

With the help of adversarial learning, the shared feature extractor $E_s$ can generate more pure task-invariant representations, which can be considered as off-the-shelf knowledge and then be used for unseen new tasks.

To test the transferability of our learned shared extractor, we also design an experiment, in which we take turns choosing 15 tasks to train our model $M_S$ with multi-task learning, then the learned shared layer are transferred to a second network $M_T$ that is used for the remaining one task. The parameters of transferred layer are **kept frozen**, and the rest of parameters of the network $M_T$ are randomly initialized.

More formally, we investigate two mechanisms towards the transferred shared extractor. As shown in Figure 4. The first one Single Channel (SC) model consists of one shared feature extractor $E_s$ from $M_S$, then the extracted representation will be sent to an output layer. By contrast, the Bi-Channel (BC) model introduces an extra LSTM layer to encode more task-specific information. To evaluate the effectiveness of our introduced adver-

| Source Tasks | Single Task | | | | Transfer Models | | | |
|---|---|---|---|---|---|---|---|---|
| | LSTM | BiLSTM | sLSTM | Avg. | SP-MTL-SC | SP-MTL-BC | ASP-MTL-SC | ASP-MTL-BC |
| $\phi$ (Books) | 20.5 | 19.0 | 18.0 | 19.2 | $17.8_{(-3.6)}$ | $16.2_{(-3.0)}$ | $16.7_{(-2.5)}$ | $13.3_{(-5.9)}$ |
| $\phi$ (Electronics) | 19.5 | 21.5 | 23.3 | 21.4 | $15.1_{(-4.5)}$ | $14.6_{(-6.8)}$ | $15.1_{(-6.3)}$ | $14.9_{(-6.5)}$ |
| $\phi$ (DVD) | 18.3 | 19.5 | 22.0 | 19.9 | $14.7_{(-3.8)}$ | $15.5_{(-4.4)}$ | $12.1_{(-7.8)}$ | $12.4_{(-7.5)}$ |
| $\phi$ (Kitchen) | 22.0 | 18.8 | 19.5 | 20.1 | $15.0_{(-3.3)}$ | $16.6_{(-3.5)}$ | $14.6_{(-5.5)}$ | $14.1_{(-6.0)}$ |
| $\phi$ (Apparel) | 16.8 | 14.0 | 16.3 | 15.7 | $14.9_{(+0.4)}$ | $12.3_{(-3.4)}$ | $11.6_{(-4.1)}$ | $13.6_{(-2.1)}$ |
| $\phi$ (Camera) | 14.8 | 14.0 | 15.0 | 14.6 | $13.1_{(-0.6)}$ | $12.1_{(-2.5)}$ | $11.6_{(-3.0)}$ | $10.3_{(-4.3)}$ |
| $\phi$ (Health) | 15.5 | 21.3 | 16.5 | 17.8 | $14.1_{(-4.9)}$ | $14.2_{(-3.6)}$ | $12.2_{(-5.6)}$ | $10.5_{(-7.3)}$ |
| $\phi$ (Music) | 23.3 | 22.8 | 23.0 | 23.0 | $19.9_{(-6.7)}$ | $17.9_{(-5.1)}$ | $16.4_{(-6.6)}$ | $18.2_{(-4.8)}$ |
| $\phi$ (Toys) | 16.8 | 15.3 | 16.8 | 16.3 | $13.8_{(-5.4)}$ | $12.2_{(-4.1)}$ | $13.0_{(-4.7)}$ | $11.2_{(-5.1)}$ |
| $\phi$ (Video) | 18.5 | 16.3 | 16.3 | 17.0 | $14.2_{(+1.7)}$ | $15.1_{(-1.9)}$ | $14.8_{(-2.2)}$ | $14.8_{(-2.2)}$ |
| $\phi$ (Baby) | 15.3 | 16.5 | 15.8 | 15.9 | $16.6_{(-3.5)}$ | $16.9_{(+1.0)}$ | $11.5_{(-4.4)}$ | $10.0_{(-5.9)}$ |
| $\phi$ (Magazines) | 10.8 | 8.5 | 12.3 | 10.5 | $10.6_{(+1.8)}$ | $10.2_{(-0.3)}$ | $8.6_{(-1.9)}$ | $9.7_{(-0.8)}$ |
| $\phi$ (Software) | 15.3 | 14.3 | 14.5 | 14.7 | $13.0_{(-1.3)}$ | $12.7_{(-2.0)}$ | $14.3_{(-0.4)}$ | $11.1_{(-3.6)}$ |
| $\phi$ (Sports) | 18.3 | 16.0 | 17.5 | 17.3 | $16.3_{(-1.2)}$ | $16.2_{(-1.1)}$ | $13.4_{(-3.9)}$ | $13.6_{(-3.7)}$ |
| $\phi$ (IMDB) | 18.3 | 15.0 | 18.5 | 17.3 | $12.4_{(-3.6)}$ | $12.8_{(-4.5)}$ | $12.5_{(-4.8)}$ | $13.3_{(-4.0)}$ |
| $\phi$ (MR) | 27.3 | 25.3 | 28.0 | 26.9 | $26.0_{(-1.4)}$ | $26.5_{(-0.4)}$ | $22.7_{(-4.2)}$ | $23.5_{(-3.4)}$ |
| AVG | 18.2 | 17.4 | 18.3 | 18.0 | $15.5_{(-2.5)}$ | $15.1_{(-2.9)}$ | $13.6_{(-4.2)}$ | $13.4_{(-4.6)}$ |

Table 3: Error rates of our models on 16 datasets against vanilla multi-task learning. $\phi$ (Books) means that we transfer the knowledge of the other 15 tasks to the target task Books.

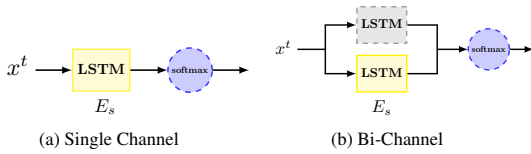

(a) Single Channel (b) Bi-Channel

Figure 4: Two transfer strategies using a pre-trained shared LSTM layer. Yellow box denotes shared feature extractor $E_s$ trained by 15 tasks.

sarial training framework, we also make a comparison with vanilla multi-task learning method.

**Results and Analysis** As shown in Table 3, we can see the shared layer from ASP-MTL achieves a better performance compared with SP-MTL. Besides, for the two kinds of transfer strategies, the Bi-Channel model performs better. The reason is that the task-specific layer introduced in the Bi-Channel model can store some private features. Overall, the results indicate that we can save the existing knowledge into a shared recurrent layer using adversarial multi-task learning, which is quite useful for a new task.

### 5.6 Visualization

To get an intuitive understanding of how the introduced orthogonality constraints worked compared with vanilla shared-private model, we design an experiment to examine the behaviors of neurons from private layer and shared layer. More concretely, we refer to $h_{tj}$ as the activation of the $j$-

neuron at time step $t$, where $t \in \{1, \dots, n\}$ and $j \in \{1, \dots, d\}$. By visualizing the hidden state $\mathbf{h}_j$ and analyzing the maximum activation, we can find what kinds of patterns the current neuron focuses on.

Figure 5 illustrates this phenomenon. Here, we randomly sample a sentence from the validation set of Baby task and analyze the changes of the predicted sentiment score at different time steps, which are obtained by SP-MTL and our proposed model. Additionally, to get more insights into how neurons in shared layer behave diversely towards different input word, we visualize the activation of two typical neurons. For the positive sentence "Five stars, my baby can fall asleep soon in the stroller", both models capture the informative pattern "Five stars" [6]. However, SP-MTL makes a wrong prediction due to misunderstanding of the word "asleep".

By contrast, our model makes a correct prediction and the reason can be inferred from the activation of Figure 5-(b), where the shared layer of SP-MTL is so sensitive that many features related to other tasks are included, such as "asleep", which misleads the final prediction. This indicates the importance of introducing adversarial learning to prevent the shared layer from being contaminated by task-specific features.

---

[6]For this case, the vanilla LSTM also give a wrong answer due to ignoring the feature "Five stars".

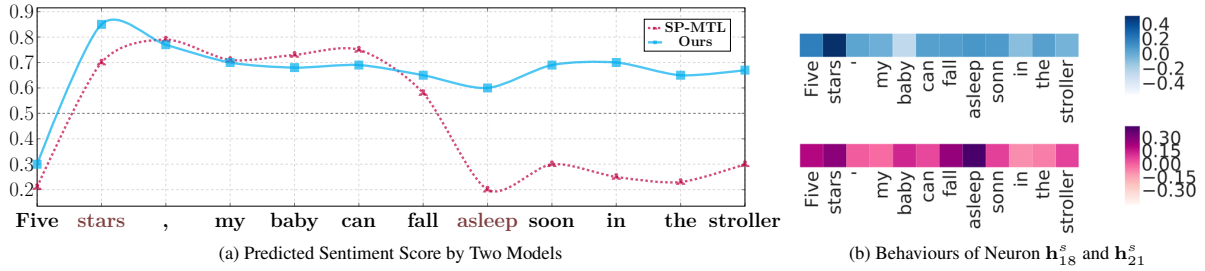

(a) Predicted Sentiment Score by Two Models

(b) Behaviours of Neuron $\mathbf{h}_{18}^s$ and $\mathbf{h}_{21}^s$

Figure 5: (a) The change of the predicted sentiment score at different time steps. Y-axis represents the sentiment score, while X-axis represents the input words in chronological order. The darker grey horizontal line gives a border between the positive and negative sentiments. (b) The blue heat map describes the behaviour of neuron $\mathbf{h}_{18}^s$ from shared layer of SP-MTL, while the purple one is used to show the behaviour of neuron $\mathbf{h}_{21}^s$, which belongs to the shared layer of our model.

| Model | Shared Layer | Task-Movie | Task-Baby |
|---|---|---|---|
| SP-MTL | good, great bad, love, simple, cut, slow, cheap, infantile | good, great, well-directed, pointless, cut, cheap, infantile | love, bad, cute, safety, mild, broken simple |
| ASP-MTL | good, great, love, bad poor | well-directed, pointless, cut, cheap, infantile | cute, safety, mild, broken simple |

Table 4: Typical patterns captured by shared layer and task-specific layer of SP-MTL and ASP-MTL models on Movie and Baby tasks.

We also list some typical patterns captured by neurons from shared layer and task-specific layer in Table 4, and we have observed that: 1) for SP-MTL, if some patterns are captured by task-specific layer, they are likely to be placed into shared space. Clearly, suppose we have many tasks to be trained jointly, the shared layer bear much pressure and must sacrifice substantial amount of capacity to capture the patterns they actually do not need. Furthermore, some typical task-invariant features also go into task-specific layer. 2) for ASP-MTL, we find the features captured by shared and task-specific layer have a small amount of intersection, which allows these two kinds of layers can work effectively.

## 6 Related Work

There are two threads of related work. One thread is multi-task learning with neural network. Neural networks based multi-task learning has been proven effective in many NLP problems (Collobert and Weston, 2008; Glorot et al., 2011; Liu et al., 2015, 2016). In most of these models, the lower layers are shared across all tasks, while top layers are task-specific. These work has potential limitation of just learning a shared space solely on sharing parameters, while our model introduce two strategies to learn the clear and non-redundant shared-private space.

Another thread of work is adversarial network. Adversarial networks have recently surfaced as a general tool measure equivalence between distributions and it has proven to be effective in a variety of tasks. Ajakan et al. (2014); Bousmalis et al. (2016) applied adverarial training to domain adaptation, aiming at transferring the knowledge of one source domain to target domain. Park and Im (2016) proposed a novel approach for multimodal representation learning which uses adversarial back-propagation concept.

Different from these models, our model aims to find task-invariant sharable information for multiple related tasks using adversarial training strategy. Moreover, we extend binary adversarial training to multi-class, which enable multiple tasks to be jointly trained.

## 7 Conclusion

In this paper, we have proposed an adversarial multi-task learning framework, in which the task-specific and task-invariant features are learned non-redundantly, therefore capturing the shared-private separation of different tasks. We have demonstrated the effectiveness of our approach by applying our model to 16 different text classification tasks. We also perform extensive qualitative analysis, deriving insights and indirectly explaining the quantitative improvements in the overall performance.

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
