# Peer review of "Adversarial Multi-task Learning for Text Classification"

_ACL 2017 — decision unknown_

[Official Review · Reviewer 1 · rating 4 · confidence 2]
soundness 5 · originality 5 · clarity 4 · impact 3 · substance 4 · appropriateness 5 · meaningful comparison 3 · presentation format Oral Presentation

This paper introduces new configurations and training objectives for neural
sequence models in a multi-task setting. As the authors describe well, the
multi-task setting is important because some tasks have shared information
and in some scenarios learning many tasks can improve overall performance.

The methods section is relatively clear and logical, and I like where it ended
up, though it could be slightly better organized. The organization that I
realized after reading is that there are two problems: 1) shared features end
up in the private feature space, and 2) private features end up in the 
shared space. There is one novel method for each problem. That organization up
front would make the methods more cohesive. In any case, they introduce one 
method that keeps task-specific features out of shared representation
(adversarial
loss) and another to keep shared features out of task-specific representations
(orthogonality constraints). My only point of confusion is the adversarial
system.
After LSTM output there is another layer, D(s^k_T, \theta_D), relying on
parameters
U and b. This output is considered a probability distribution which is compared
against the actual. This means it is possible it will just learn U and b that
effectively mask task-specific information from  the LSTM outputs, and doesn't 
seem like it can guarantee task-specific information is removed.

Before I read the evaluation section I wrote down what I hoped the experiments
would look like and it did most of it. This is an interesting idea and there
are 
a lot more experiments one can imagine but I think here they have the basics
to show the validity of their methods. It would be helpful to have best known
results on these tasks.

My primary concern with this paper is the lack of deeper motivation for the 
approach. I think it is easy to understand that in a totally shared model
there will be problems due to conflicts in feature space. The extension to 
partially shared features seems like a reaction to that issue -- one would 
expect that the useful shared information is in the shared latent space and 
each task-specific space would learn features for that space. Maybe this works
and maybe it doesn't, but the logic is clear to me. In contrast, the authors
seem to start from the assumption that this "shared-private" model has this
issue. I expected the argument flow to be 1) Fully-shared obviously has this
problem; 2) shared-private seems to address this; 3) in practice shared-private
does not fully address this issue for reasons a,b,c.; 4) we introduce a method
that more effectively constrains the spaces.
Table 4 helped me to partially understand what's going wrong with
shared-private
and what your methods do; some terms are _usually_ one connotation
or another, and that general trend can probably get them into the shared
feature
space. This simple explanation, an example, and a more logical argument flow
would help the introduction and make this a really nice reading paper.

Finally, I think this research ties into some other uncited MTL work [1],
which does deep hierarchical MTL - supervised POS tagging at a lower level,
chunking
at the next level up, ccg tagging higher, etc. They then discuss at the end
some of the qualities that make MTL possible and conclude that MTL only works
"when tasks are sufficiently similar." The ASP-MTL paper made me think of this
previous work because potentially this model could learn what sufficiently
similar is -- i.e., if two tasks are not sufficiently similar the shared model
would learn nothing and it would fall back to learning two independent systems,
as compared to a shared-private model baseline that might overfit and perform
poorly.

[1]
@inproceedings{sogaard2016deep,
  title={Deep multi-task learning with low level tasks supervised at lower
layers},
  author={S{\o}gaard, Anders and Goldberg, Yoav},
  booktitle={Proceedings of the 54th Annual Meeting of the Association for
Computational Linguistics},
  volume={2},
  pages={231--235},
  year={2016},
  organization={Association for Computational Linguistics}
}

[Official Review · Reviewer 2 · rating 4 · confidence 3]
soundness 5 · originality 5 · clarity 4 · impact 3 · substance 5 · appropriateness 5 · meaningful comparison 3 · presentation format Oral Presentation

# Paper summary

This paper presents a method for learning well-partitioned shared and
task-specific feature spaces for LSTM text classifiers. Multiclass adversarial
training encourages shared space representations from which a discriminative
classifier cannot identify the task source (and are thus generic). The models
evaluates are a fully-shared, shared-private and adversarial shared-private --
the lattermost ASP model is one of the main contributions. They also use
orthogonality constraints to help reward shared and private spaces that are
distinct. The ASP model has lower error rate than single-task and other
multi-task neural models. They also experiment with a task-level cross
validation to explore whether the shared representation can transfer across
tasks, and it seems to favourably. Finally, there is some analysis of shared
layer activations suggesting that the ASP model is not being misled by strong
weights learned on a specific (inappropriate) task.

# Review summary

Good ideas, well expressed and tested. Some minor comments.

# Strengths

* This is a nice set of ideas working well together. I particularly like the
focus on explicitly trying to create useful shared representations. These have
been quite successful in the CV community, but it appears that one needs to
work quite hard to create them for NLP.
* Sections 2, 3 and 4 are very clearly expressed.
* The task-level cross-validation in Section 5.5 is a good way to evaluate the
transfer.
* There is an implementation and data.

# Weaknesses

* There are a few minor typographic and phrasing errors. Individually, these
are fine, but there are enough of them to warrant fixing:
** l:84 the “infantile cart” is slightly odd -- was this a real example
from the data?
** l:233 “are different in” -> “differ in”
** l:341 “working adversarially towards” -> “working against” or
“competing with”?
** l:434 “two matrics” -> “two matrices”
** l:445 “are hyperparameter” -> “are hyperparameters”
** Section 6 has a number of number agreement errors
(l:745/746/765/766/767/770/784) and should be closely re-edited.
** The shading on the final row of Tables 2 and 3 prints strangely…
* There is mention of unlabelled data in Table 1 and semi-supervised learning
in Section 4.2, but I didn’t see any results on these experiments. Were they
omitted, or have I misunderstood?
* The error rate differences are promising in Tables 2 and 3, but statistical
significance testing would help make them really convincing. Especially between
SP-MLT and ASP-MTL results to highlight the benefit of adversarial training. It
should be pretty straightforward to adapt the non-parametric approximate
randomisation test (see
http://www.lr.pi.titech.ac.jp/~takamura/pubs/randtest.pdf for promising notes a
reference to the Chinchor paper) to produce these.
* The colours are inconsistent in the caption of Figure 5 (b). In 5 (a), blue
is used for “Ours”, but this seems to have swapped for 5 (b). This is worth
checking, or I may have misunderstood the caption.

# General Discussion

* I wonder if there’s some connection with regularisation here, as the effect
of the adversarial training with orthogonal training is to help limit the
shared feature space. It might be worth drawing that connection to other
regularisation literature.